# HBV seroprevalence and liver fibrosis status among population born before national immunization in Southern Thailand: Findings from a health check-up program

**Supinya Sono**[1,2☯], **Jirayu Sae-Chan**[3☯], **Apichat Kaewdech**[2,3], **Naichaya Chamroonkul**[3]*, **Pimsiri Sripongpun**[2,3]*

1 Division of Family and Preventive Medicine, Faculty of Medicine, Prince of Songkla University, Hat Yai, Thailand, 2 Srivejchavat Premium Center, Songklanagarind Hospital, Hat Yai, Thailand, 3 Division of Internal Medicine, Faculty of Medicine, Prince of Songkla University, Hat Yai, Thailand

☯ These authors contributed equally to this work.
* spimsiri@medicine.psu.ac.th (PS); naichaya@gmail.com (NC)

**Data Availability Statement:** Data cannot be shared publicly because of ethical restrictions involving patient information. Data are available

## Abstract

### Background

Hepatitis B virus (HBV) infection is the leading cause of liver-related death worldwide, particularly in Asia. Patients with either current or past HBV infection are at risk of cirrhosis and hepatocellular carcinoma (HCC). Here, we investigated the HBV seroprevalence in residents of southern Thailand born before the national vaccination program.

### Methods

A cross-sectional study of individuals born before the nationwide HBV vaccination program who sought check-up programs which included HBV serology and abdominal ultrasound at a tertiary care hospital in southern Thailand from 2019 to 2020 was conducted. HBV serology, cirrhosis and liver fibrosis status (determined by ultrasonography and FIB-4), and clinical notes regarding management following HBsAg+ detection were obtained.

### Results

Of 1,690 eligible individuals, the overall prevalence of HBsAg+ and HBsAg-antiHBc+, indicating current and past HBV infections, were 2.9% and 27.8%, respectively. Among current HBV patients, 87.8% were unaware of their infection. Cirrhosis was found in 8.2%, 1.1%, and 0.3% of patients with current, past, and no HBV infection, respectively (p<0.001). One-fourth of past HBV patients had FIB-4≥1.45, which indicated indeterminate or significant liver fibrosis, which may increase the risk of HCC.

### Conclusion

The prevalence of HBsAg+ in Southern Thailand was 2.9%, and that of past infection (HBsAg-antiHBc+) was 27.8%. Patients with current and past HBV infection have an increased risk of cirrhosis and significant liver fibrosis. Most current HBV patients were

from the Division of Digital Innovation and Data Analytics (DIDA), Faculty of Medicine, Prince of Songkhla University after appropriate protocol submission to the institution's office of Human Research Ethics Committee (HREC) (contact via medpsu.ec@gmail.com, referring to the REC 64-079-9-1) for researchers who meet the criteria for access to confidential data.

**Funding:** The author(s) received no specific funding for this work.

**Competing interests:** The authors have declared that no competing interests exist.

unaware of their infection. Identifying patients with current and past HBV infection who are at risk for HCC and linking them to a cascade of care is necessary to reduce the burden of HBV-related liver diseases in Thailand.

## Introduction

Hepatitis B virus (HBV) infection is a major cause of chronic hepatitis, liver cirrhosis, and hepatocellular carcinoma (HCC), resulting in approximately 820,000 deaths annually [1, 2]. Worldwide, it is estimated that 296 million people are living with HBV, however, only 10.5% of all HBV infected patients are aware of their infection status, and only 22% of them receive treatment [1].

The prevalence of HBV infection in children has significantly declined since nationwide HBV vaccination was officially implemented in 1992 in Thailand as a part of the Expanded Programme on Immunization (EPI). A recent study in Thailand published in 2019 [3] reported that the prevalence of HBV infection in Thailand ranged between 0.1% and 5.99% depending on the age group [4], similar to an earlier study by Ott et al. [5], which reported that the prevalence of HBV infection in Southeast Asia was 6%.

Most of the previous studies in Thailand mainly focused on the seroprevalence of HBsAg [3–7], but the prevalence of hepatitis B core antibody (anti-HBc) has been far less explored, and data on the liver fibrosis status of those who had hepatitis B have been reported to a lesser degree. In the current era of immunologic/biologic therapies for the treatment of various diseases, being HBsAg negative but anti-HBc positive (HBsAg-antiHBc+), which indicates 'past HBV infection', is of clinical importance, as it poses a risk of HBV reactivation in patients receiving those novel therapies [8]. Therefore, our study aimed to assess the seroprevalence of HBsAg and anti-HBc, and liver fibrosis status in the population of Southern Thailand using data from individuals who voluntarily participated in health check-up programs.

## Material and methods

### Setting and study population

This is a cross-sectional study of individuals who attended a health check-up program at Sri-vejchavat Premium Center, Songklanagarind Hospital, the only super-tertiary care center in Southern Thailand, between January 1st, 2019 and December 31st, 2020. We only included individuals born before 1992 (the year EPI for HBV immunization was initiated nationwide in Thailand) who chose the check-up program options packages E and F, which include measurement of HBsAg, anti-HBc, and anti-HBs, as well as abdominal ultrasound. Hepatitis C virus antibody (anti-HCV) and human immunodeficiency virus antibody (anti-HIV) were not included in the general checkup programs. Non-Thai nationals and patients residing outside of the 14 southern Thai provinces were excluded. The study was approved by the Office of the Human Research Ethics Committee (HREC), Faculty of Medicine, Prince of Songkhla University (REC 64-079-9-1), and was conducted in accordance with the principles and standards of the Declaration of Helsinki and Good Clinical Practice guidelines.

### Data collection

We retrospectively retrieved all data from the medical records via our institutional Hospital Information System. The variables included age, sex, province of residence, serological

markers of HBV including HBsAg, anti-HBs, and anti-HBc, liver biochemistry (aspartate aminotransferase (AST), alanine aminotransferase (ALT), and alkaline phosphatase (ALP)), platelet count, liver ultrasound results, and clinicians' notes on whether the patients were transferred to a gastroenterology/hepatology specialist if they were positive for HBsAg were obtained. As the data were retrospectively retrieved, informed consent of the patients was waived under the HREC approval.

### Fibrosis stage evaluation

The diagnosis of cirrhosis was made based on liver ultrasonographic results reported by radiologists. Typically, radiologists at our center report the ultrasound findings in two parts: a description of the findings and a summary. The description part may vary, but the summary part of the liver is usually uniformly reported as either normal, liver parenchymal disease, or cirrhosis. Only patients with a diagnosis of cirrhosis in the summary part of the ultrasound reports were considered to have cirrhosis. Noninvasive fibrosis scores of fibrosis-4 (FIB-4) and AST-to-platelet ratio (APRI) were calculated using the standard formulas [9, 10] in order to evaluate any fibrosis stages milder than cirrhosis (fibrosis stage F0-3 by METAVIR) using the cut-off values for determining significant fibrosis (F2 or higher) recommended by the World Health Organization (WHO) as follows: FIB-4 <1.45, a low threshold to exclude significant fibrosis; FIB-4 of 1.45–3.25, an indeterminate significant fibrosis status; and FIB-4 >3.25, a high cut-off level to rule in significant fibrosis [11].

### Statistical analysis

Based on an expected HBsAg seroprevalence of 6% according to a previous study [3] with a precision margin of error of 0.02, a calculated sample size of at least 542 patients would be sufficient for the prevalence analysis. In this analysis, descriptive statistics are presented as mean ± standard deviation (SD) or median (interquartile range [IQR]) for continuous data, according to the distribution of the data, and as percentages for categorical data. The ninety-five percent confidence interval (95%CI) was also presented for HBV seroprevalence. Comparisons of characteristics between patients according to HBsAg and anti-HBc Ag status (HBsAg positive, positive anti-HBc but not HBsAg, and negative for both HBsAg and anti-HBc groups) were analyzed using chi-square or Fisher's exact test for categorical variables, while ANOVA or Kruskal-Wallis test was used for continuous variables. The proportions of patients with cirrhosis and significant liver fibrosis are presented as numbers (percentages), and chi-squared or Fisher's exact tests were performed to compare the differences in fibrosis status between the groups. All tests were two-tailed, and a p-value <0.05 was considered statistically significant. All statistical analyses were performed using R program [12], version 4.1.0.

### Results

Between January 1, 2019, and December 31, 2020, a total of 1,743 distinct patients underwent health check-ups with one of the aforementioned packages at our center. Twenty-six non-Thai nationals and 27 patients residing in provinces outside southern Thailand were excluded. A total of 1,690 patients were therefore eligible and included in the analyses.

Overall, the median age of the eligible patients was 55 (IQR: 46,61) years, with roughly equal proportions of men and women (48.6% vs 51.4%, respectively); 44.4% of the patients lived in Songkhla province, where our center is located, followed by Yala (11.2%), Nakhon Si Thammarat (8.5%), Phattalung (8.3%), and Pattani (7.6%).

## Hepatitis B serology findings

Seroprevalences of HBsAg positivity (HBsAg+), positivity for anti-HBc but not HBsAg (HBsAg-antiHBc+), and negativity for both HBsAg and anti-HBc (HBsAg-antiHBc-) were found in 49 (2.9%, [95%CI: 2.2–3.8%]), 469 (27.8%, [95%CI: 25.7–29.9%]), and 1,172 (69.3%, [95%CI: 67.1–71.5%]) patients, respectively. All HBsAg+ patients (100%) were positive for anti-HBc, indicating a true HBV infection status. While 82.5% (95%CI: 78.8–85.7%) of patients with HBsAg-antiHBc+ were also anti-HBs positive, reflecting that past HBV infection is the vast majority status of the patients with HBsAg-antiHBc+, whereas the remaining 17.5% (95%CI: 14.3–21.2%) who were negative for anti-HBs could have past, resolved infection but could also have falsely positive anti-HBc results, or low- level chronic HBV infection, or resolution of acute infection. The demographic data and characteristics of the patients according to their HBV serological status are shown in Table 1. Patients with HBsAg+ were younger (median age of 52 [IQR: 46, 57] years) than patients with HBsAg-antiHBc+ (median age 59 [IQR: 53, 63] years), and HBsAg-antiHBc- patients (median age 54 [IQR: 44,60] years), respectively. Higher mean AST levels and lower platelet counts were observed in HBsAg+ patients compared to the other two groups.

The patients were stratified into five age groups: 28–40 years (n = 240), 41–50 years (n = 355), 51–60 years (n = 640), 61–70 years (n = 373), and > 70 years (n = 82). The prevalence of HBsAg-positive and anti-HBc positive in each group are shown in Fig 1A and 1B.

**Table 1. Characteristics of all eligible patients according to HBV serology status.**

| Characteristics | Total (N = 1,690) | HBsAg-antiHBc- (N = 1,172) | HBsAg-antiHBc+ (N = 469) | HBsAg+ (N = 49) | P-value |
|---|---|---|---|---|---|
| AGE, years; median (IQR) | 55 (46,61) | 54 (44,60) | 59 (53,63) | 52 (46,57) | < 0.001 |
| Male sex; n (%) | 822 (48.6) | 522 (44.5) | 268 (57.1) | 32 (65.3) | < 0.001 |
| Race-Ethnicity | | | | | 0.988 |
| • Thai | 1604 (94.9) | 1113 (95) | 445 (94.9) | 46 (93.9) | |
| • Thai-Chinese | 4 (0.2) | 3 (0.3) | 1 (0.2) | 0 (0) | |
| • Thai-Muslim | 82 (4.9) | 56 (4.8) | 23 (4.9) | 3 (6.1) | |
| Province of residence; n (%) | | | | | 0.007 |
| Chumphon | 3 (0.2) | 2 (0.2) | 0 (0) | 1 (2) | |
| Krabi | 43 (2.5) | 30 (2.6) | 13 (2.8) | 0 (0) | |
| Nakhon Si Thammarat | 143 (8.5) | 102 (8.7) | 39 (8.3) | 2 (4.1) | |
| Narathiwat | 79 (4.7) | 50 (4.3) | 24 (5.1) | 5 (10.2) | |
| Pattani | 129 (7.6) | 87 (7.4) | 35 (7.5) | 7 (14.3) | |
| Phangnga | 4 (0.2) | 4 (0.3) | 0 (0) | 0 (0) | |
| Phattalung | 141 (8.3) | 105 (9) | 31 (6.6) | 5 (10.2) | |
| Phuket | 25 (1.5) | 20 (1.7) | 5 (1.1) | 0 (0) | |
| Ranong | 3 (0.2) | 2 (0.2) | 1 (0.2) | 0 (0) | |
| Satun | 82 (4.9) | 55 (4.7) | 23 (4.9) | 4 (8.2) | |
| Songkhla | 751 (44.4) | 530 (45.2) | 202 (43.1) | 19 (38.8) | |
| Surat Thani | 27 (1.6) | 11 (0.9) | 16 (3.4) | 0 (0) | |
| Trang | 71 (4.2) | 44 (3.8) | 23 (4.9) | 4 (8.2) | |
| Yala | 189 (11.2) | 130 (11.1) | 57 (12.2) | 2 (4.1) | |
| AST (U/L); median (IQR) | 21 (18,26) | 21 (18,25) | 22 (19,27) | 22 (19,26) | 0.013 |
| ALT (U/L); median (IQR) | 21 (15,30) | 21 (15,30) | 21 (16,30) | 22 (17,27) | 0.664 |
| WBC (×10$^9$/L); median (IQR) | 6.2 (5.3,7.3) | 6.2 (5.3,7.3) | 6.2 (5.3,7.5) | 5.6 (5.1,6.5) | 0.017 |
| Hemoglobin (g/dL); mean (SD) | 13.8 (1.5) | 13.8 (1.5) | 14 (1.5) | 14.4 (1.3) | 0.002 |
| Platelets (×10$^9$/L); median (IQR) | 261 (224,298) | 265 (228,302.8) | 253.5 (214.8,289) | 235 (195,265) | < 0.001 |

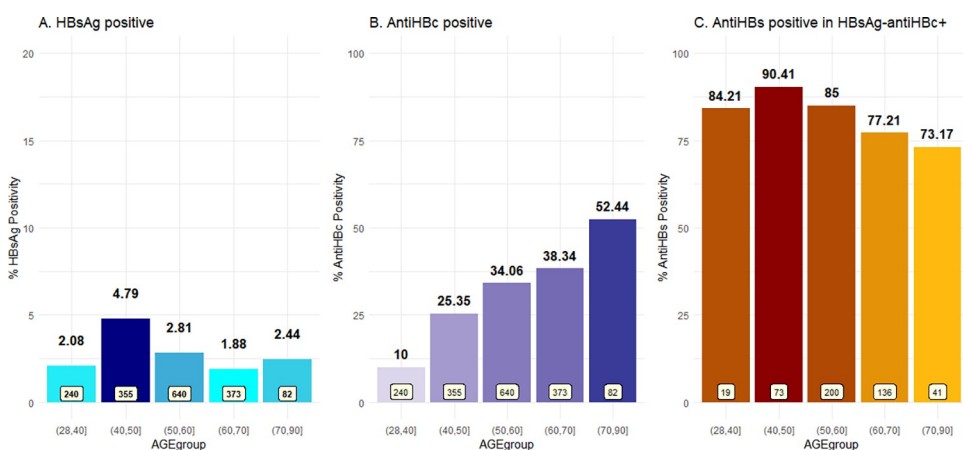

**Fig 1.** Prevalence of (A) overall HBsAg positivity, (B) anti-HBc positivity according to age group in all eligible patients, and (C) anti-HBs positive in HBsAg- antiHBc+ patients.

While HBsAg seroprevalence varied from 1.88 to 4.79% among the groups, the seroprevalence of anti-HBc increased with age, from 10% in patients aged under 40 years to 52.4% in patients aged over 70 years. Fig 1C shows the prevalence of anti-HBs positivity among those who were HBsAg-antiHBc+.

Although there were individuals from all 14 provinces in Southern Thailand presented for check-ups at our center, patients who resided in Songkhla and the eight provinces in the immediate vicinity accounted for over 90% of the entire eligible patients in this study. The seroprevalence of HBsAg in these provinces is shown in Fig 2, while there were too few patients from Chumphon, Ranong, Surat Thani, Pangnga, and Phuket, patients from these provinces are not presented in this figure.

## Liver fibrosis status

Table 2 and Fig 3 demonstrate the fibrosis stage assessment in eligible patients using APRI and FIB-4 scores and liver ultrasonographic findings. Cirrhosis (fibrosis stage 4) was observed in 4 of the 49 HBsAg+ patients (8.2%); this rate was significantly higher than in HBsAg-antiHBc + patients (1.1%) and patients without HBV infection (0.3%) (p<0.001). Focal hepatic lesion (s) compatible with hepatic nodule(s) were found in 2.3–3.8% of patients, but no HCC was identified in patients with nodule size ≥ 1 cm who subsequently underwent dynamic contrast imaging. We used the noninvasive biomarkers APRI and FIB-4 to evaluate lesser degrees of fibrosis (fibrosis stages 0–3), which cannot be evaluated by ultrasound. Although statistically significant levels were reached, the median APRI and FIB-4 scores were not clearly distinguishable among the three groups (Table 2). When the WHO cutoffs [11] were applied to identify patients with significant fibrosis (fibrosis ≥F2), the proportions of patients classified as no, indeterminate, and having significant fibrosis differed among the three groups (Fig 3A). Eighty-seven percent of the patients who did not have HBV infection had no significant liver fibrosis, but more than 20% of HBsAg+ and HBsAg-antiHBc+ patients were considered to have indeterminate or significant fibrosis.

## Patient management following detection of HBsAg positivity

Of the 49 patients who were positive for HBsAg, designating current HBV infection status, only 6 (12.2%) were aware of their infection and were being regularly followed up with a

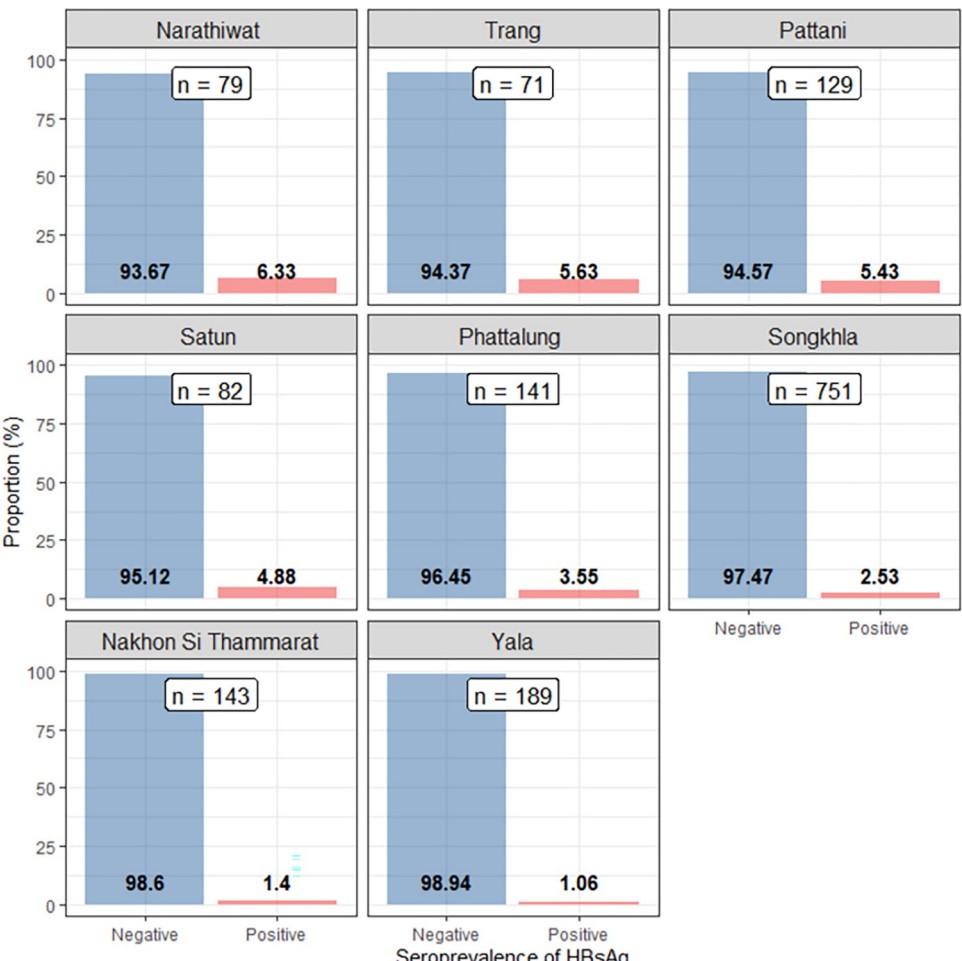

**Fig 2. Prevalence of HBsAg positive patients identified on health check-up according to province (Songkhla and provinces in direct vicinity).**

**Table 2. Noninvasive fibrosis score and liver ultrasonographic findings of patients according to HBV serology status.**

| | HBsAg-antiHBc-(N = 1,172) | HBsAg-antiHBc+ (N = 469) | HBsAg+ (N = 49) | P-value |
|---|---|---|---|---|
| APRI; median (IQR) | 0.20 (0.16,0.26) | 0.22 (0.17,0.30) | 0.23 (0.20,0.29) | < 0.001 |
| FIB-4; median (IQR) | 0.92 (0.67, 1.20) | 1.08 (0.83, 1.44) | 1.00 (0.82, 1.34) | < 0.001 |
| FIB-4 score category*, n (%) | | | | < 0.001 |
| • <1.45 | 1013 (87.1) | 355 (75.9) | 39 (79.6) | |
| • 1.45–3.25 | 139 (12) | 97 (20.7) | 8 (16.3) | |
| • >3.25 | 11 (0.9) | 16 (3.4) | 2 (4.1) | |
| Ultrasonographic findings; n (%) | | | | |
| • Cirrhosis | 4 (0.3) | 5 (1.1) | 4 (8.2) | < 0.001 |
| • Steatosis | 574 (49) | 234 (49.9) | 20 (40.8) | 0.481 |
| • Nodule(s) | 45 (3.8) | 11 (2.3) | 61 (3.6) | 0.015 |

*FIB-4 score category to determine the risk of F2 fibrosis or higher according to the WHO recommendation [11].

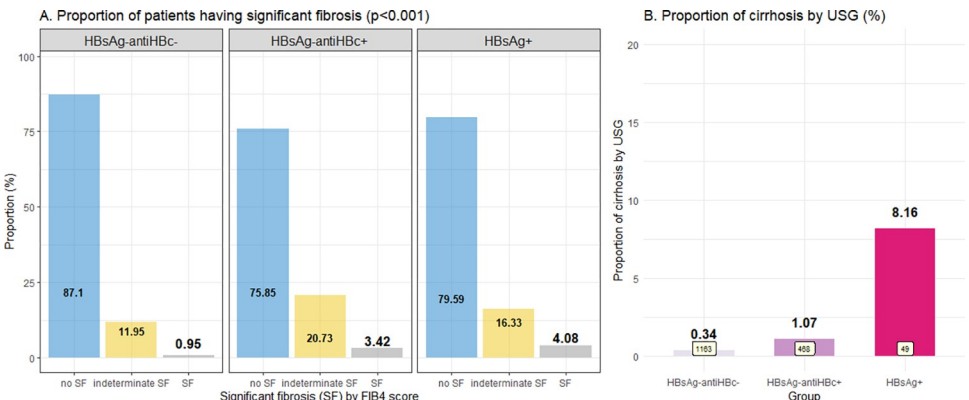

**Fig 3. Fibrosis status of the patients according to age group.** (A) Significant fibrosis by FIB-4 score, and (B) Cirrhosis by ultrasonography.

gastroenterology/hepatology specialist before participating in the health check-up program. Of the remaining 43 patients in whom HBV infection was first diagnosed at the time of check-up, 95.3% were advised by their primary check-up physician to undergo further investigation and follow-up with a specialist regarding their HBV infection.

## Discussion

This study represents the seroprevalence of HBV in the general population undergoing health checkups in Southern Thailand in recent years. Although data regarding the seroprevalence of HBsAg in Thailand had been studied in a substantial number of publications earlier, few studies have included the seroprevalence of anti-HBc, and even fewer studies that examined correlations between HBV serology and liver fibrosis. Our study data were extracted from a database where all HBsAg, anti-HBc, anti-HBs data, as well as liver biochemistry, and liver ultrasonography results were included.

We found that the prevalence of HBsAg+ individuals in Southern Thailand born before EPI implementation was 2.9% [95%CI: 2.2–3.8%], ranging from 1.88 to 4.79% among different age groups. This is numerically lower than the most recent report on the general population in Thailand born before the EPI program of 4.3% [13], but similar to a 2007 report on the first blood donors in Yala, a province in Southern Thailand (2.7%) [14], and family/replacement blood donors at Songklanagarind Hospital (our center) in 2001–2003 (2.58%) [15]. Differences in prevalence rates may exist among the different regions of Thailand.

Despite the difference in the prevalence of HBsAg+ patients (current HBV infection) between our study and a previous study in the general population in Thailand [13], the anti-HBc+ prevalence was similar (27.8% and 29.6%, respectively). Likewise, we found that the seroprevalence of anti-HBc increased with age, which was similar to a recent study of older adults [3] from Khon Kaen, a province in northeastern Thailand, in which approximately half of the patients were positive for anti-HBc. Further, 82% of HBsAg- antiHBc+ patients were positive for anti-HBs, indicating that most had past HBV infection, thus increasing the risk of cirrhosis and the development of HCC.

When comparing patients with current HBV infection (HBsAg+), past HBV infection (HBsAg-antiHBc+), and no HBV infection (HBsAg-antiHBc-), 8.2% of our patients with current HBV infection already had cirrhosis detected by ultrasound, which was significantly higher than the 1.1% with a past HBV infection, and 0.3% in the patients with no HBV infection, respectively (p<0.001). Furthermore, using the FIB-4 score as a non-invasive marker of

liver fibrosis which has been validated in HBV patients [16], an increased risk of having significant liver fibrosis (F2 or higher) was observed not only in patients with current HBV infection (20.41%) but also in patients with past HBV infection (24.15%) compared to patients without HBV infection (12.9%). In addition to being a determinant of liver fibrosis status, FIB-4 has also been reported to be an independent predictor of HCC occurrence in patients with HBV. The FIB-4 cutoff for an increased risk of HCC development varied among studies, from $\geq 1.29$ [17], $\geq 2.4$ [18], and $\geq 3.25$ [19].

In terms of clinical significance, as the WHO has set the goal for viral hepatitis elimination by 2030 [20], one of the most important challenges in achieving this goal is to identify patients with infection [21]. In the present study, almost 90% of current HBV-infected patients detected through the health check-up program were unaware of their infection status, moreover, it was estimated that in 2016, only 5% of HBV-infected individuals in Thailand had been diagnosed [22]. Additionally, not all patients newly diagnosed with current HBV infection in our cohort were advised to undergo further evaluation and follow-up. These findings represent opportunities for improvement in HBV care in Thailand; HBV screening should be implemented broadly, and all patients with HBV should be taken up in the cascade of care.

Patients with past HBV infection, on the other hand, were thought to be at a low risk of long-term liver-related complications. Nonetheless, they were not as safe as patients without HBV infection, as mentioned earlier, and 1.07% of them already had cirrhosis. Those with a high FIB-4 score might be at a higher risk of developing HCC in the future, even though the FIB-4 cutoff for determining HCC risk in patients with past HBV infection is yet to be determined. Patients with past HBV infection (HBsAg-antiHBc+) were also recently demonstrated to be associated with a higher risk of HCC in patients with non-alcoholic fatty liver disease [23] and other chronic liver diseases [24]. Moreover, a previous study reported that positive anti-HBc was observed in >75% of non-HBV, non-HCV HCC new cases in an HBV-endemic area [25], suggesting that past HBV infection was potentially linked to HCC development, and those with past HBV infection may benefit from HCC surveillance [26, 27].

Furthermore, in the era of new biologic/targeted therapies for various diseases, including cancers, autoimmune disorders, etc., the risk of HBV reactivation following those novel treatments should be considered [28]. While the reactivation risk in HBsAg+ patients undergoing immunosuppressive treatments is well established and prophylactic antivirals have been recommended for more than a decade [29], data regarding reactivation in patients with past HBV infection treated novel biologic treatments are non-negligible [30, 31]. Almost 30% of the patients in our study exhibited a status indicating past HBV infection, and prophylactic antivirals or close monitoring with liver function tests and/or HBV DNA should be considered if they were to receive any immunotherapy in the future.

The present study is the first to explore the seroprevalence of HBsAg, anti-HBc, and anti-HBs and their correlation with liver biochemistry and liver fibrosis status in Thai patients. The strengths of our study are that all patients had liver biochemistry and ultrasound data, in which we could illustrate the clinical picture of the liver fibrosis status of all patient categories: current HBV infection, past HBV infection, and no HBV infection. This provides informative epidemiological data linking the current seroprevalence and the clinical status of patients in Thailand.

We acknowledge some limitations of our study, primarily that the study design did not use stratified random sampling to acquire participants; therefore, it may not represent the actual prevalence at the population level. However, we included all patients from all 14 provinces in Southern Thailand who sought health checkups at our center during the study period, which is likely to represent patients who are at general risk and unaware of their HBV status in the most recent timeframe. The patients included in this study accounted for only one-third of all

patients seeking health checkups at our center during the study period. It is therefore important to note that, as the different check-up packages come with different costs, those who chose to get packages that included HBV serology and ultrasonography may have a higher socioeconomic status than those who chose health checkups at a lower cost. Although a recent meta-analysis demonstrated that low-income status may increase the risk of HBV seroprevalence, the sensitivity analysis of patients residing in Asia in that meta-analysis showed no significant association between low-income status and HBV seroprevalence [32]. A prior study in Thailand also showed that HBsAg seroprevalence was not statistically significantly higher in individuals with an income <10,000 baht/month (approximately 300 USD) than in those with a higher monthly income [33]. Therefore, the prevalence of current HBV infection in our cohort may be underestimated, as only those who voluntarily selected the higher-cost packages were included. However, we believe that this difference should not be statistically significant. Another limitation is that anti-HCV, anti-HIV, and history of alcohol consumption, which may also contribute to liver disease and liver fibrosis, were not systematically tested and obtained. And HBV DNA data for patients who were HBsAg-positive were not entirely available.

## Conclusion

In summary, our study found that the prevalence of current HBV infection among patients residing in southern Thailand was 2.9%. The overall prevalence of past HBV infection was 27.8%, and the prevalence of HBsAg-antiHBc+ increased with age, with the highest prevalence (52.4%) observed in patients over 70 years of age. Those with current and past HBV infection had higher proportions of cirrhosis and significant liver fibrosis than those without HBV infection. Almost 90% of patients with current HBV infection were unaware of their HBV status. Identifying patients with current HBV infection or those with past infection who were at risk of cirrhosis and HCC development and linking them to care is required to decrease the burden of HBV-related liver diseases in Thailand.

## Acknowledgments

We would like to thank the Division of Digital Innovation and Data Analytics (DIDA), Faculty of Medicine, Prince of Songkla University, for facilitating data extraction from the Hospital Information System.

## Author Contributions

**Conceptualization:** Apichat Kaewdech, Naichaya Chamroonkul, Pimsiri Sripongpun.

**Data curation:** Supinya Sono, Jirayu Sae-Chan, Pimsiri Sripongpun.

**Formal analysis:** Pimsiri Sripongpun.

**Methodology:** Supinya Sono, Pimsiri Sripongpun.

**Project administration:** Supinya Sono, Pimsiri Sripongpun.

**Supervision:** Apichat Kaewdech, Naichaya Chamroonkul, Pimsiri Sripongpun.

**Visualization:** Apichat Kaewdech, Pimsiri Sripongpun.

**Writing – original draft:** Supinya Sono, Jirayu Sae-Chan.

**Writing – review & editing:** Apichat Kaewdech, Naichaya Chamroonkul, Pimsiri Sripongpun.

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
