## [Decision Letter · Decision Letter 0]

14 Mar 2022

PONE-D-22-05319HBV Seroprevalence and Liver Fibrosis Status Among Population Born Before National Immunization in Southern Thailand: Findings from Health Check-Up ProgramPLOS ONE

Dear Dr. Sripongupun,

Thank you for submitting your manuscript to PLOS ONE. After careful consideration, we feel that it has merit but does not fully meet PLOS ONE’s publication criteria as it currently stands. Therefore, we invite you to submit a revised version of the manuscript that addresses the points raised during the review process.

We look forward to receiving your revised manuscript.

Kind regards,

Jason T. Blackard, PhD

Academic Editor

PLOS ONE

Journal Requirements:

2. We note that Figure 2 in your submission contain [map/satellite] images which may be copyrighted. All PLOS content is published under the Creative Commons Attribution License (CC BY 4.0), which means that the manuscript, images, and Supporting Information files will be freely available online, and any third party is permitted to access, download, copy, distribute, and use these materials in any way, even commercially, with proper attribution. For these reasons, we cannot publish previously copyrighted maps or satellite images created using proprietary data, such as Google software (Google Maps, Street View, and Earth). For more information, see our copyright guidelines: http://journals.plos.org/plosone/s/licenses-and-copyright.

Additional Editor Comments:

This is a cross-sectional study of HBV and liver fibrosis in southern Thailand.

This is a well-written epidemiologic study.  While the findings are not entirely unexpected, the methods are appropriate and the results informative.  Additional clarifications would strengthen the manuscript further, including:

Was HCV or HIV serostatus considered?

Additional study limitations include the lack of HBV DNA quantification.

In Table 2, it would be helpful to stratify FIB-4 scores into low / high (>1.45 etc).

Reviewers' comments:

Reviewer's Responses to Questions

**Comments to the Author**

1. Is the manuscript technically sound, and do the data support the conclusions?

Reviewer #1: Partly

Reviewer #2: Yes

2. Has the statistical analysis been performed appropriately and rigorously? 

Reviewer #1: I Don't Know

Reviewer #2: Yes

3. Have the authors made all data underlying the findings in their manuscript fully available?

Reviewer #1: Yes

Reviewer #2: No

4. Is the manuscript presented in an intelligible fashion and written in standard English?

Reviewer #1: No

Reviewer #2: Yes

5. Review Comments to the Author

Reviewer #1: The English language should be revised and improved throughout the manuscript.

Abstract (and Manuscript):

The Authors should explain how liver cirrhosis was defined by ultrasound, and which morphologic features were considered suggestive of liver cirrhosis.

The Authors state that HBsAg-antiHBc+ indicate past HBV infections. However, even though in most cases the HBsAg-anti-HBc+ pattern does indicate past, resolved infection, this pattern can also indicate false-positive anti-HBc, low level chronic HBV infection and even resolving acute infection. These possibilities should be mentioned for the 17.5% of patients in this group who did not have concomitant anti-HBs positivity.

Abstract:

Lines 40-41: ''One-fourth of past HBV patients had FIB-4>1.45 which correlates with significant fibrosis and increased HCC risk''.

FIB-4 < 1.45 can rule-out advanced fibrosis, but only at > 3.25 can rule-in advanced fibrosis; this should be clarified. In addition, only HBsAg carriers with FIB-4 > 2.2-2.4 have an increased HCC risk, and this also should be rectified.

Discussion:

Lines 188-190: ''And eighty-two percent of HBsAg-antiHBc+ patients were also positive for anti-HBs, most of them had had a past HBV infection, thus increasing the risk of cirrhosis and the development of HCC''. The Authors should clarify on the basis of which evidence past HBV infection increases the risk of cirrhosis and the development of HCC.

Lines 213-214: ''and HCC surveillance is still recommended even in patients with HBV who turned to HBsAg seroconversion (antiHBc+, antiHBs+)''. The Authors refer to a 2011 article for this statement. They should provide much more recent evidence that surveillance is recommended nowadays.

Lines 235-243 repeat what already stated and should be removed.

Table 1: Marital status, creatinine and ALP should be removed, as irrelevant.

Table 2. The Authors should provide an explanation as to why the ultrasound revealed such a high number of patients with

steatosis.

The data on gallstones are not relevant and should be removed.

The Authors should explain what ‘’nodules’’ mean. Does it mean surface nodularity? If so, why is there a

discrepancy with liver cirrhosis? And why 61 cases in 49 patients are indicated (this is clearly wrong, and the 3.6

percentage is also wrong).

Reviewer #2: This well written article summarizes a cross-sectional analysis of electronic medical records to understand the frequency of hepatitis B virus (HBV), current and past infections, and HBV’s association with advanced liver diagnostic tests usage and their findings. A total of 1690 Thai citizens living in the south, who had a qualifying health examination at one medical center, were included. The findings are consistent with prior studies in Thailand and regionally, both on HBsAg+ prevalence and the lack of awareness of the infection. The authors argue that providing more information on those with HBsAg- but anti-HBc+ is valuable as this group could have their HBV infection reactivated by some therapies. While the study’s findings are not particularly novel, they have value in providing a current understanding of HBV status as elimination efforts continue.

Major comment

The key issue that needs to be addressed is who was eligible for the study. Specifically, who selected packages E and F for the health assessment? What proportion of all health assessments at the medical center are E or F; what proportion of the population have health assessments elsewhere, etc. Could the estimates be biased away from the null (i.e., slight overestimates)? If so, then several edits would be in order: take the word “population” out of the title, mention this eligibility in the abstract, qualify the key finding and add information to the limitations.

Details

Abbreviations: it would be helpful to define the terms used in the abstract and the first usage in the manuscript.

Confidence intervals for prevalence estimates would be helpful to readers.

Consider using the term “proportion” in place of “rate.”

170 “health check-up in Thailand”

Add ‘southern’ Thailand

231 “However, we included all patients from all 14 provinces in 232 Southern Thailand who sought health check-ups at our center during the study period,”

Actually, only those asking for specific packages.

Is there any history of alcohol documented in the medical records? It seems that this may be a limitation given alcohol’s association with liver disease.

6. PLOS authors have the option to publish the peer review history of their article (what does this mean?). If published, this will include your full peer review and any attached files.

Reviewer #1: No

Reviewer #2: No

---

## [Author Response · Author response to Decision Letter 0]

4 May 2022

RESPONSE:- Thank you, we have edited the format of the entire manuscript according to the PLOS ONE’s style.

2. We note that Figure 2 in your submission contain [map/satellite] images which may be copyrighted. All PLOS content is published under the Creative Commons Attribution License (CC BY 4.0), which means that the manuscript, images, and Supporting Information files will be freely available online, and any third party is permitted to access, download, copy, distribute, and use these materials in any way, even commercially, with proper attribution. For these reasons, we cannot publish previously copyrighted maps or satellite images created using proprietary data, such as Google software (Google Maps, Street View, and Earth). For more information, see our copyright guidelines: http://journals.plos.org/plosone/s/licenses-and-copyright.

RESPONSE:- - Thank you. We have removed figure 2 showing map figure, replaced it with simple figure instead.

RESPONSE:- We have reviewed the reference and change the reference style according to the journal’s style

Additional Editor Comments:

This is a cross-sectional study of HBV and liver fibrosis in southern Thailand.

This is a well-written epidemiologic study. While the findings are not entirely unexpected, the methods are appropriate and the results informative. Additional clarifications would strengthen the manuscript further, including:

Was HCV or HIV serostatus considered?

Additional study limitations include the lack of HBV DNA quantification.

RESPONSE- Thank you, editor, we have added the limitation statement regarding HCV, HIV, and HBV DNA tests in the discussion part.

In Table 2, it would be helpful to stratify FIB-4 scores into low / high (>1.45 etc).

RESPONSE:-The proportions of each FIB-4 category had been added to table 2 already.

Reviewers' comments:

5. Review Comments to the Author

Reviewer #1: The English language should be revised and improved throughout the manuscript.

RESPONSE: - We have carefully re-read the entire manuscript and made language editing and polishing thoroughly. Thank you for pointing out.

Abstract (and Manuscript):

The Authors should explain how liver cirrhosis was defined by ultrasound, and which morphologic features were considered suggestive of liver cirrhosis.

RESPONSE- The diagnosis of cirrhosis using ultrasound findings was explained in a further detail in material and methods part.

The Authors state that HBsAg-antiHBc+ indicate past HBV infections. However, even though in most cases the HBsAg-anti-HBc+ pattern does indicate past, resolved infection, this pattern can also indicate false-positive anti-HBc, low level chronic HBV infection and even resolving acute infection. These possibilities should be mentioned for the 17.5% of patients in this group who did not have concomitant anti-HBs positivity.

RESPONSE:- Thank you, we have added the statement regarding this issue onto the results part as “All HBsAg+ patients (100%) were also positive for anti-HBc, indicating true HBV infection status. While 82.5% (95%CI: 78.8-85.7%) of patients with HBsAg-antiHBc+ also had anti-HBs positivity, reflecting that past HBV infection is the vast majority status of the patients with HBsAg-antiHBc+, whereas the remaining 17.5% (95%CI: 14.3-21.2%) who were negative for anti-HBs might be from either low level of anti-HBs, low HBsAg level, or false-positive anti-HBc.”

Abstract:

Lines 40-41: ''One-fourth of past HBV patients had FIB-4>1.45 which correlates with significant fibrosis and increased HCC risk''.

FIB-4 < 1.45 can rule-out advanced fibrosis, but only at > 3.25 can rule-in advanced fibrosis; this should be clarified. In addition, only HBsAg carriers with FIB-4 > 2.2-2.4 have an increased HCC risk, and this also should be rectified.

RESPONSE:- Thank you, we made a better clarification regarding FIB-4 strata on the presence of fibrosis level as well as for HCC risk. The correction had been made to the material and methods part as “the cut-off values for determining significant fibrosis (F2 or higher) recommended by the World Health Organization (WHO) as follow: FIB-4 of <1.45 is a low threshold to exclude significant fibrosis, FIB-4 of 1.45-3.25 demonstrates indeterminate significant fibrosis status, and FIB-4 >3.25 is the high cut-off level to rule-in significant fibrosis.[11]”, and for HCC risk in the discussion part as “Furthermore, using the FIB-4 score as a noninvasive marker of liver fibrosis which has been validated in HBV patients[16], an increased risk of having significant liver fibrosis (F2 or higher) were observed not only in patients with current HBV infection (20.41%), but also in patients with past HBV infection (24.15%) when compared with patients without HBV infection (12.9%). Not only being a determinant of liver fibrosis status, FIB-4 had also been reported to be an independent predictor of HCC occurrence in patients with HBV. The FIB-4 cut-offs for an increased risk of HCC development varied among studies, from >1.29[17], >2.4[18], to >3.25[19]”.

Discussion:

Lines 188-190: ''And eighty-two percent of HBsAg-antiHBc+ patients were also positive for anti-HBs, most of them had had a past HBV infection, thus increasing the risk of cirrhosis and the development of HCC''. The Authors should clarify on the basis of which evidence past HBV infection increases the risk of cirrhosis and the development of HCC.

Lines 213-214: ''and HCC surveillance is still recommended even in patients with HBV who turned to HBsAg seroconversion (antiHBc+, antiHBs+)''. The Authors refer to a 2011 article for this statement. They should provide much more recent evidence that surveillance is recommended nowadays.

RESPONSE:- For the discussion part, we have made a clarification on the role of past HBV infection on HCC development and added more recent references as “ Patients with past HBV infection, on the other hand, were thought to be at a low risk of long-term liver-related complications. Nonetheless, they were not as safe as patients without HBV infection, as mentioned earlier, 1.07% of them already had cirrhosis. And those with a high FIB-4 score might be at a higher risk of developing HCC in the future, even though the FIB-4 cutoff for determining HCC risk in patients with past HBV infection is yet to be determined. Patients with past HBV infection (HBsAg-antiHBc+) were also recently demonstrated to be associated with a higher risk of HCC in patients with non-alcoholic fatty liver disease[23], and other chronic liver diseases.[24] Moreover, there was a study reported that positive anti-HBc was observed in >75% of non-HBV, non-HCV HCC new cases in an HBV-endemic area,[25] suggesting that past HBV infection potentially linked to HCC development and those with past HBV infection might benefit from HCC surveillance. [26, 27]”

Table 1: Marital status, creatinine and ALP should be removed, as irrelevant.

RESPONSE:- We have removed those parameters as suggested

The data on gallstones are not relevant and should be removed.

The Authors should explain what ‘’nodules’’ mean. Does it mean surface nodularity? If so, why is there a discrepancy with liver cirrhosis? And why 61 cases in 49 patients are indicated (this is clearly wrong, and the 3.6 percentage is also wrong).

RESPONSE:- The hepatic nodule(s) means focal liver lesions not nodularity of the liver, we have clarified this on the result part. And removal of gallstones findings from the table has been done.

 

Reviewer #2: This well written article summarizes a cross-sectional analysis of electronic medical records to understand the frequency of hepatitis B virus (HBV), current and past infections, and HBV’s association with advanced liver diagnostic tests usage and their findings. A total of 1690 Thai citizens living in the south, who had a qualifying health examination at one medical center, were included. The findings are consistent with prior studies in Thailand and regionally, both on HBsAg+ prevalence and the lack of awareness of the infection. The authors argue that providing more information on those with HBsAg- but anti-HBc+ is valuable as this group could have their HBV infection reactivated by some therapies. While the study’s findings are not particularly novel, they have value in providing a current understanding of HBV status as elimination efforts continue.

Major comment

The key issue that needs to be addressed is who was eligible for the study. Specifically, who selected packages E and F for the health assessment? What proportion of all health assessments at the medical center are E or F; what proportion of the population have health assessments elsewhere, etc. Could the estimates be biased away from the null (i.e., slight overestimates)? If so, then several edits would be in order: take the word “population” out of the title, mention this eligibility in the abstract, qualify the key finding and add information to the limitations.

RESPONSE:- Thank you for the valuable comment, ones who chose package E and F were accounted for one-third of all health check-up individuals. We have added the statement regarding this issue in the limitation part as “In addition, as the different check-up packages comes with different costs, those who chose to get packages which included HBV serology and ultrasonography may have a higher socioeconomic status than those who chose health check-up packages at a lower cost (the patients included in this study were accounted for one-third of all patients sought for health check-up at our center during the study period). Although a recent meta-analysis demonstrated that the low-income status may increase risk of HBV seroprevalence, the sensitivity analysis of patients residing in Asia in that meta-analysis showed no significant association between the low-income status and HBV seroprevalence.[32] A prior study in Thailand also showed that the HBsAg seroprevalence was not statistically significant higher in individuals who had income <10,000 baht/month (approximately 300 USD) compared to those who had higher monthly income.[33] Therefore, the prevalence of current HBV infection in our cohort might be underestimated as only those who voluntarily selected the higher cost packages were included, but we think it should not be considerably statistically significant.” 

Details

Abbreviations: it would be helpful to define the terms used in the abstract and the first usage in the manuscript.

Confidence intervals for prevalence estimates would be helpful to readers.

Consider using the term “proportion” in place of “rate.”

170 “health check-up in Thailand” Add ‘southern’ Thailand

231 “However, we included all patients from all 14 provinces in 232 Southern Thailand who sought health check-ups at our center during the study period,” Actually, only those asking for specific packages.

RESPONSE:- Thank you. We have made corrections throughout the manuscript as per the reviewer’s suggestion.

Is there any history of alcohol documented in the medical records? It seems that this may be a limitation given alcohol’s association with liver disease.

RESPONSE:- Alcohol intake was not universally recorded; we have added this a one of the limitations of our study in the discussion part. Thank you.

---

## [Decision Letter · Decision Letter 1]

26 May 2022

PONE-D-22-05319R1HBV Seroprevalence and Liver Fibrosis Status Among Population Born Before National Immunization in Southern Thailand: Findings from Health Check-Up ProgramPLOS ONE

Dear Dr. Sripongpun,

Thank you for submitting your manuscript to PLOS ONE. After careful consideration, we feel that it has merit but does not fully meet PLOS ONE’s publication criteria as it currently stands. Therefore, we invite you to submit a revised version of the manuscript that addresses the points raised during the review process.

Please address the minor revisions raised by the reviewer and have your manuscript reviewed carefully by a native English speaker and/or a professional editing service prior to resubmission.

We look forward to receiving your revised manuscript.

Kind regards,

Jason T. Blackard, PhD

Academic Editor

PLOS ONE

Journal Requirements:

Additional Editor Comments:

Please address the minor revisions raised by the reviewer and have your manuscript reviewed carefully by a native English speaker and/or a professional editing service prior to resubmission.

Reviewers' comments:

Reviewer's Responses to Questions

**Comments to the Author**

1. If the authors have adequately addressed your comments raised in a previous round of review and you feel that this manuscript is now acceptable for publication, you may indicate that here to bypass the “Comments to the Author” section, enter your conflict of interest statement in the “Confidential to Editor” section, and submit your "Accept" recommendation.

Reviewer #1: (No Response)

2. Is the manuscript technically sound, and do the data support the conclusions?

Reviewer #1: Yes

3. Has the statistical analysis been performed appropriately and rigorously? 

Reviewer #1: I Don't Know

4. Have the authors made all data underlying the findings in their manuscript fully available?

Reviewer #1: Yes

5. Is the manuscript presented in an intelligible fashion and written in standard English?

Reviewer #1: No

6. Review Comments to the Author

Reviewer #1: The manuscript has been considerably improved. However, the English is not entirely up to standard yet.

In addition, the statement "....whereas the remaining 17.5% (95%CI: 14.3-21.2%) who were negative for anti-HBs might be from either low level of anti-HBs, low HBsAg level, or false-positive anti-HBc" is not ideal. It should read : ""....whereas the remaining 17.5% (95%CI: 14.3-21.2%) who were negative for anti-HBs could have past, resolved infection but could also have falsely positive anti-HBc results or low level chronic HBV infection or resolving acute infection".

7. PLOS authors have the option to publish the peer review history of their article (what does this mean?). If published, this will include your full peer review and any attached files.

Reviewer #1: No

---

## [Author Response · Author response to Decision Letter 1]

7 Jun 2022

Responses to reviewers:

Additional Editor Comments:

Please address the minor revisions raised by the reviewer and have your manuscript reviewed carefully by a native English speaker and/or a professional editing service prior to resubmission.

RESPONSE: Thank you for your comments, we have revised according to the reviewer’s comments. And we also have a proofread by a professional editing service, the certificate of English editing is also submitted in the platform.

Reviewers' comments:

Reviewer's Responses to Questions

Comments to the Author

5. Is the manuscript presented in an intelligible fashion and written in standard English?

Reviewer #1: No

RESPONSE: Thank you, in this revised manuscript, we have a proofread by a professional editing service, the certificate of English editing is also submitted in the platform.

6. Review Comments to the Author

Reviewer #1: The manuscript has been considerably improved. However, the English is not entirely up to standard yet.

In addition, the statement "....whereas the remaining 17.5% (95%CI: 14.3-21.2%) who were negative for anti-HBs might be from either low level of anti-HBs, low HBsAg level, or false-positive anti-HBc" is not ideal. It should read : ""....whereas the remaining 17.5% (95%CI: 14.3-21.2%) who were negative for anti-HBs could have past, resolved infection but could also have falsely positive anti-HBc results or low level chronic HBV infection or resolving acute infection".

RESPONSE: Thank you, the correction has been made accordingly as follows: “whereas the remaining 17.5% (95%CI: 14.3-21.2%) who were negative for anti-HBs could have past, resolved infection but could also have falsely positive anti-HBc results, or low- level chronic HBV infection, or resolution of acute infection” (Results part: line 202-205).

---

## [Editor Report · Decision Letter 2]

12 Jun 2022

HBV Seroprevalence and Liver Fibrosis Status Among Population Born Before National Immunization in Southern Thailand: Findings from Health Check-Up Program

PONE-D-22-05319R2

Dear Dr. Sripongun,

We’re pleased to inform you that your manuscript has been judged scientifically suitable for publication and will be formally accepted for publication once it meets all outstanding technical requirements.

Kind regards,

Jason T. Blackard, PhD

Academic Editor

PLOS ONE

Additional Editor Comments (optional):

None

Reviewers' comments:

None

---

## [Editor Report · Acceptance letter]

16 Jun 2022

PONE-D-22-05319R2 

HBV seroprevalence and liver fibrosis status among population born before national immunization in Southern Thailand: Findings from a health check-up program 

Dear Dr. Sripongpun:

I'm pleased to inform you that your manuscript has been deemed suitable for publication in PLOS ONE. Congratulations! Your manuscript is now with our production department. 

Kind regards, 

on behalf of

Dr. Jason T. Blackard 

Academic Editor

PLOS ONE